# Posterior Collapse and
# Latent Variable Non-identifiability

**Yixin Wang**
University of Michigan
yixinw@umich.edu

**David M. Blei**
Columbia University
david.blei@columbia.edu

**John P. Cunningham**
Columbia University
jpc2181@columbia.edu

## Abstract

Variational autoencoders model high-dimensional data by positing low-dimensional latent variables that are mapped through a flexible distribution parametrized by a neural network. Unfortunately, variational autoencoders often suffer from posterior collapse: the posterior of the latent variables is equal to its prior, rendering the variational autoencoder useless as a means to produce meaningful representations. Existing approaches to posterior collapse often attribute it to the use of neural networks or optimization issues due to variational approximation. In this paper, we consider posterior collapse as a problem of latent variable non-identifiability. We prove that the posterior collapses if and only if the latent variables are non-identifiable in the generative model. This fact implies that posterior collapse is not a phenomenon specific to the use of flexible distributions or approximate inference. Rather, it can occur in classical probabilistic models even with exact inference, which we also demonstrate. Based on these results, we propose a class of latent-identifiable variational autoencoders, deep generative models which enforce identifiability without sacrificing flexibility. This model class resolves the problem of latent variable non-identifiability by leveraging bijective Brenier maps and parameterizing them with input convex neural networks, without special variational inference objectives or optimization tricks. Across synthetic and real datasets, latent-identifiable variational autoencoders outperform existing methods in mitigating posterior collapse and providing meaningful representations of the data.

## 1 Introduction

Variational autoencoders (VAE) are powerful generative models for high-dimensional data [28, 46]. Their key idea is to combine the inference principles of probabilistic modeling with the flexibility of neural networks. In a VAE, each datapoint is independently generated by a low-dimensional latent variable drawn from a prior, then mapped to a flexible distribution parametrized by a neural network.

Unfortunately, VAE often suffer from posterior collapse, an important and widely studied phenomenon where the posterior of the latent variables is equal to prior [6, 8, 38, 62]. This phenomenon is also known as latent variable collapse, KL vanishing, and over-pruning. Posterior collapse renders the VAE useless to produce meaningful representations, in so much as its per-datapoint latent variables all have the exact same posterior.

Posterior collapse is commonly observed in the VAE whose generative model is highly flexible, leading to the common speculation that posterior collapse occurs because VAE involve flexible neural networks in the generative model [11], or because it uses variational inference [59]. Based on these hypotheses, many of the proposed strategies for mitigating posterior collapse thus focus on modifying the variational inference objective (e.g. [44]), designing special optimization schemes for variational inference in VAE (e.g. [18, 25, 32]), or limiting the capacity of the generative model (e.g. [6, 16, 60].)

35th Conference on Neural Information Processing Systems (NeurIPS 2021).

In this paper, we consider posterior collapse as a problem of latent variable non-identifiability. We prove that posterior collapse occurs if and only if the latent variable is non-identifiable in the generative model, which loosely means the likelihood function does not depend on the latent variable [40, 42, 56]. Below, we formally establish this equivalence by appealing to recent results in Bayesian non-identifiability [40, 42, 43, 49, 58].

More broadly, the relationship between posterior collapse and latent variable non-identifiability implies that posterior collapse is not a phenomenon specific to the use of neural networks or variational inference. Rather, it can also occur in classical probabilistic models fitted with exact inference methods, such as Gaussian mixture models and probabilistic principal component analysis (PPCA). This relationship also leads to a new perspective on existing methods for avoiding posterior collapse, such as the delta-VAE [44] or the $\beta$-VAE [19]. These methods heuristically adjust the approximate inference procedure embedded in the optimization of the model parameters. Though originally motivated by the goal of patching the variational objective, the results here suggest that these adjustments are useful because they help avoid parameters at which the latent variable is non-identifiable and, consequently, avoid posterior collapse.

The relationship between posterior collapse and non-identifiability points to a direct solution to the problem: we must make the latent variable identifiable. To this end, we propose latent-identifiable VAE, a class of VAE that is as flexible as classical VAE while also being identifiable. Latent-identifiable VAE resolves the latent variable non-identifiability by leveraging Brenier maps [36, 39] and parameterizing them with input-convex neural networks [2, 35]. Inference on identifiable VAE uses the standard variational inference objective, without special modifications or optimization tricks. Across synthetic and real datasets, we show that identifiable VAE mitigates posterior collapse without sacrificing fidelity to the data.

**Related work.** Existing approaches to avoiding posterior collapse often modify the variational inference objective, design new initialization or optimization schemes for VAE, or add neural network links between each data point and their latent variables [1, 3, 6, 8, 12, 15, 16, 17, 18, 21, 25, 27, 32, 34, 38, 44, 50, 51, 52, 55, 61, 62, 63]. Several recent papers also attempt to provide explanations for posterior collapse. Chen et al. [8] explains how the inexact variational approximation can lead to inefficiency of coding in VAE, which could lead to posterior collapse due to a form of information preference. Dai et al. [11] argues that posterior collapse can be partially attributed to the local optima in training VAE with deep neural networks. Lucas et al. [33] shows that posterior collapse is not specific to the variational inference training objective; absent a variational approximation, the log marginal likelihood of PPCA has bad local optima that can lead to posterior collapse. Yacoby et al. [59] discusses how variational approximation can select an undesirable generative model when the generative model parameters are non-identifiable. In contrast to these works, we consider posterior collapse solely as a problem of latent variable non-identifiability, and not of optimization, variational approximations, or neural networks per se. We use this result to propose the identifiable VAE as a way to directly avoid posterior collapse.

Outside VAE, latent variable identifiability in probabilistic models has long been studied in the statistics literature [40, 42, 42, 43, 49, 56, 58]. More recently, Betancourt [5] studies the effect of latent variable identifiability on Bayesian computation for Gaussian mixtures. Khemakhem et al. [23, 24] propose to resolve the non-identifiability in deep generative models by appealing to auxiliary data. Kumar & Poole [29] study how the variational family can help resolve the non-identifiability of VAE. These works address the identifiability issue for a different goal: they develop identifiability conditions for different subsets of VAE, aiming for recovering true causal factors of the data and improving disentanglement or out-of-distribution generalization. Related to these papers, we demonstrate posterior collapse as an additional way that the concept of identifiability, though classical, can be instrumental in modern probabilistic modeling. Considering identifiability leads to new solutions to posterior collapse.

**Contributions.** We prove that posterior collapse occurs if and only if the latent variable in the generative model is non-identifiable. We then propose latent-identifiable VAE, a class of VAE that are as flexible as classical VAE but have latent variables that are provably identifiable. Across synthetic and real datasets, we demonstrate that latent-identifiable VAE mitigates posterior collapse without modifying VAE objectives or applying special optimization tricks.

## 2 Posterior collapse and latent variable non-identifiability

Consider a dataset $\boldsymbol{x} = (x_1, \ldots, x_n)$; each datapoint is $m$-dimensional. Positing $n$ latent variables $\boldsymbol{z} = (z_1, \ldots, z_n)$, a variational autoencoder (VAE) assumes that each datapoint $x_i$ is generated by a $K$-dimensional latent variable $z_i$:

$$z_i \sim p(z_i), \qquad x_i \,|\, z_i \sim p(x_i \,|\, z_i\,; \theta) = \mathrm{EF}(x_i \,|\, f_\theta(z_i)), \tag{1}$$

where $x_i$ follows an exponential family distribution with parameters $f_\theta(z_i)$; $f_\theta$ parameterizes the conditional likelihood. In a deep generative model $f_\theta$ is a parameterized neural network. Classical probabilistic models like Gaussian mixture model [45] and probabilistic PCA [10, 47, 48, 54] are also special cases of Eq. 1.

To fit the model, VAE optimizes the parameters $\theta$ by maximizing a variational approximation of the log marginal likelihood. After finding an optimal $\hat{\theta}$, we can form a representation of the data using the approximate posterior $q_{\hat{\phi}}(z \,|\, x)$ with variational parameters $\hat{\phi}$ or its expectation $\mathbb{E}_{q_{\hat{\phi}}(z|x)}[z \,|\, x]$.

Note that here we abstract away computational considerations and consider the ideal case where the variational approximation is exact. This choice is sensible: if the exact posterior suffers from posterior collapse then so will the approximate posterior (a variational approximation cannot "uncollapse" a collapsed posterior). That said we also note that there exist in practice situations where variational inference alone can lead to posterior collapse. A notable example is when the variational approximating family is overly restrictive: it is then possible to have non-collapsing exact posteriors but collapsing approximate posteriors.

### 2.1 Posterior collapse ⇔ Latent variable non-identifiability

We first define posterior collapse and latent variable non-identifiability, then proving their connection.

**Definition 1** (Posterior collapse [6, 8, 38, 62]). *Given a probability model $p(\boldsymbol{x}, \boldsymbol{z}\,; \theta)$, a parameter value $\theta = \hat{\theta}$, and a dataset $\boldsymbol{x} = (x_1, \ldots, x_n)$, the posterior of the latent variables $\boldsymbol{z}$ collapses if*

$$p(\boldsymbol{z} \,|\, \boldsymbol{x}\,; \hat{\theta}) = p(\boldsymbol{z}). \tag{2}$$

The posterior collapse phenomenon can occur in a variety of probabilistic models and with different latent variables. When the probability model is a VAE, it only has local latent variables $\boldsymbol{z} = (z_1, \ldots, z_n)$, and Eq. 2 is equivalent to the common definition of posterior collapse $p(z_i \,|\, x_i\,; \hat{\theta}) = p(z_i)$ for all $i$ [12, 17, 33, 44]. Posterior collapse has also been observed in Gaussian mixture models [5]; the posterior of the latent mixture weights resembles their prior when the number of mixture components in the model is larger than that of the data generating process. Regardless of the model, when posterior collapse occurs, it prevents the latent variable from providing meaningful summary of the dataset.

**Definition 2** (Latent variable non-identifiability [42, 56]). *Given a likelihood function $p(\boldsymbol{x} \,|\, \boldsymbol{z}\,; \theta)$, a parameter value $\theta = \hat{\theta}$, and a dataset $\boldsymbol{x} = (x_1, \ldots, x_n)$, the latent variable $\boldsymbol{z}$ is non-identifiable if*

$$p(\boldsymbol{x} \,|\, \boldsymbol{z} = \tilde{\boldsymbol{z}}'\,; \hat{\theta}) = p(\boldsymbol{x} \,|\, \boldsymbol{z} = \tilde{\boldsymbol{z}}\,; \hat{\theta}) \qquad \forall \tilde{\boldsymbol{z}}', \tilde{\boldsymbol{z}} \in \mathcal{Z}, \tag{3}$$

*where $\mathcal{Z}$ denotes the domain of $\boldsymbol{z}$, and $\tilde{\boldsymbol{z}}', \tilde{\boldsymbol{z}}$ refer to two arbitrary values the latent variable $\boldsymbol{z}$ can take. As a consequence, for any prior $p(z)$ on z, we have the conditional likelihood equal to the marginal $p(\boldsymbol{x} \,|\, \boldsymbol{z} = \tilde{\boldsymbol{z}}\,; \hat{\theta}) = \int p(\boldsymbol{x} \,|\, \boldsymbol{z}\,; \hat{\theta}) p(\boldsymbol{z}) \mathrm{d}\boldsymbol{z} = p(\boldsymbol{x}\,; \hat{\theta}) \quad \forall \tilde{\boldsymbol{z}} \in \mathcal{Z}$.*

Definition 2 says a latent variable $\boldsymbol{z}$ is non-identifiable when the likelihood of the dataset $\boldsymbol{x}$ does not depend on $\boldsymbol{z}$. It is also known as practical non-identifiability [42, 56] and is closely related to the definition of $\boldsymbol{z}$ being conditionally non-identifiable (or conditionally uninformative) given $\hat{\theta}$ [40, 42, 43, 49, 58]. To enforce latent variable identifiability, it is sufficient to ensure that the likelihood $p(\boldsymbol{x} \,|\, \boldsymbol{z}, \theta)$ is an injective (a.k.a. one-to-one) function of $\boldsymbol{z}$ for all $\theta$. If this condition holds then

$$\tilde{\boldsymbol{z}}' \neq \tilde{\boldsymbol{z}} \qquad \Rightarrow \qquad p(\boldsymbol{x} \,|\, \boldsymbol{z} = \tilde{\boldsymbol{z}}'\,; \hat{\theta}) \neq p(\boldsymbol{x} \,|\, \boldsymbol{z} = \tilde{\boldsymbol{z}}\,; \hat{\theta}). \tag{4}$$

Note that latent variable non-identifiability only requires Eq. 3 be true for a given dataset $\boldsymbol{x}$ and parameter value $\hat{\theta}$. Thus a latent variable may be identifiable in a model given one dataset but not another, and at one $\theta$ but not another. See examples in Appendix A.

Latent variable identifiability (Definition 2) [42, 56] differs from model identifiability [41], a related notion that has also been cited as a contributing factor to posterior collapse [59]. Latent variable

identifiability is a weaker requirement: it only requires the latent variable $\boldsymbol{z}$ be identifiable at a particular parameter value $\theta = \hat{\theta}$, while model identifiability requires both $\boldsymbol{z}$ and $\theta$ be identifiable.

We now establish the equivalence between posterior collapse and latent variable non-identifiability.

**Theorem 1** (Latent variable non-identifiability ⇔ Posterior collapse). *Consider a probability model $p(\boldsymbol{x}, \boldsymbol{z}; \theta)$, a dataset $\boldsymbol{x}$, and a parameter value $\theta = \hat{\theta}$. The local latent variables $\boldsymbol{z}$ are non-identifiable at $\hat{\theta}$ if and only if the posterior of the latent variable $\boldsymbol{z}$ collapses, $p(\boldsymbol{z} | \boldsymbol{x}) = p(\boldsymbol{z})$.*

*Proof.* To prove that non-identifiability implies posterior collapse, note that, by Bayes rule,

$$p(\boldsymbol{z} | \boldsymbol{x}; \hat{\theta}) \propto p(\boldsymbol{z}) p(\boldsymbol{x} | \boldsymbol{z}; \hat{\theta}) = p(\boldsymbol{z}) p(\boldsymbol{x}; \hat{\theta}) \propto p(\boldsymbol{z}), \tag{5}$$

where the middle equality is due to the definition of latent variable non-identifiability. It implies $p(\boldsymbol{z} | \boldsymbol{x}; \hat{\theta}) = p(\boldsymbol{z})$ as both are densities. To prove that posterior collapse implies latent variable non-identifiability, we again invoke Bayes rule. Posterior collapse implies that $p(\boldsymbol{z}) = p(\boldsymbol{z} | \boldsymbol{x}; \hat{\theta}) \propto p(\boldsymbol{z}) \cdot p(\boldsymbol{x} | \boldsymbol{z}; \hat{\theta})$, which further implies that $p(\boldsymbol{x} | \boldsymbol{z}; \hat{\theta})$ is constant in $\boldsymbol{z}$. If $p(\boldsymbol{x} | \boldsymbol{z}; \hat{\theta})$ nontrivially depends on $\boldsymbol{z}$, then $p(\boldsymbol{z})$ must be different from $p(\boldsymbol{z}) p(\boldsymbol{x} | \boldsymbol{z}; \hat{\theta})$ as a function of $\boldsymbol{z}$. ☐

The proof of Theorem 1 is straightforward, but Theorem 1 has an important implication. It shows that the problem of posterior collapse mainly arises from the model and the data, rather than from inference or optimization. If the maximum likelihood parameters $\hat{\theta}$ of the VAE renders the latent variable $z$ non-identifiable, then we will observe posterior collapse. Theorem 1 also clarifies why posteriors may change from non-collapsed to collapsed (and back) while fitting a VAE. When fitting a VAE, Some parameter iterates may lead to posterior collapse; others may not.

Theorem 1 points to why existing approaches can help mitigate posterior collapse. Consider the $\beta$-VAE [19], the VAE lagging encoder [18], and the semi-amortized VAE [25]. Though motivated by other perspectives, these methods modify the optimization objectives or algorithms of VAE to avoid parameter values $\theta$ at which the latent variable is non-identifiable. The resulting posterior may not collapse, though the optimal parameters for these algorithms no longer approximates the maximum likelihood estimate.

Theorem 1 can also help us understand posterior collapse observed in practice, which manifests as the phenomenon that the posterior is approximately (as opposed to exactly) equal to the prior, $p(\boldsymbol{z} | \boldsymbol{x}; \hat{\theta}) \approx p(\boldsymbol{z})$. In several empirical studies of VAE (e.g. [12, 18, 25]), we observe that the Kullback-Leibler (KL) divergence between the prior and posterior is close to zero but not exactly zero, a property that stems from the likelihood $p(\boldsymbol{x} | \boldsymbol{z})$ being nearly constant in the latents $\boldsymbol{z}$. In these cases, Theorem 1 provides the intuition that the latent variable is nearly non-identifiable , $p(\boldsymbol{x} | \tilde{\boldsymbol{z}}') \approx p(\boldsymbol{x} | \tilde{\boldsymbol{z}}), \forall \tilde{\boldsymbol{z}}, \tilde{\boldsymbol{z}}'$ and so Eq. 2 holds approximately.

## 2.2 Examples of latent variable non-identifiability and posterior collapse

We illustrate Theorem 1 with three examples. Here we discuss the example of Gaussian mixture VAE (GMVAE). See Appendix A for probabilistic principal component analysis (PPCA) and Gaussian mixture model (GMM).

The GMVAE [13, 51] is the following model:

$$p(z_i) = \text{Categorical}(1/K), \quad p(w_i | z_i; \mu, \Sigma) = \mathcal{N}(\mu_{z_i}, \Sigma_{z_i}), \quad p(x_i | w_i; f, \sigma) = \mathcal{N}(f(w_i), \sigma^2 \cdot I_m),$$

where $\mu_k$'s are $d$-dimensional, $\Sigma_k$ are $d \times d$-dimensional, and the parameters are $\theta = (\mu, \Sigma, f, \sigma^2)$. Suppose the function $f$ is fully flexible; thus $f(w_i)$ can capture any distribution of the data. The latent variable of interest is the categorical $\boldsymbol{z} = (z_1, \ldots, z_n)$. If its posterior collapses, then $p(z_i = k | \boldsymbol{x}) = 1/K$ for all $k = 1, \ldots, K$.

Consider fitting a GMVAE model with $K = 2$ to a dataset of 5,000 samples. This dataset is drawn from a GMVAE also with $K = 2$ well-separated clusters; there is no model misspecification. A GMVAE is typically fit by optimizing the maximum log marginal likelihood $\hat{\theta} = \text{argmax}_\theta \log p(\boldsymbol{x} | \theta)$. Note there may be multiple values of $\theta$ that achieve the global optimum of this function.

We focus on two likelihood maximizers. One provides latent variable identifiability and the posterior of $z_i$ does not collapse. The other does not provide identifiablity; the posterior collapses.

1. The first likelihood-maximizing parameter $\hat{\theta}_1$ is the truth; the distribution of the $K$ fitted clusters correspond to the $K$ data-generating clusters. Given this parameter, the latent variable $z_i$ is identifiable because the $K$ data-generating clusters are different; different cluster memberships $z_i$ must result in different likelihoods $p(x_i|z_i;\hat{\theta}_1)$. The posterior of $z_i$ does not collapse.

2. In the second likelihood-maximizing parameter $\hat{\theta}_2$, however, all $K$ fitted clusters share the same distribution, each of which is equal to the marginal distribution of the data. Specifically, $(\mu_k^*, \Sigma_k^*) = (0, I_d)$ for all $k$, and each fitted cluster is a mixture of the $K$ original data generating clusters, i.e., the marginal. At this parameter value, the model is still able to fully capture the mixture distribution of the data. However, all the $K$ mixture components are the same, and thus the latent variable $z_i$ is non-identifiable; different cluster membership $z_i$ do not result in different likelihoods $p(x_i|z_i;\hat{\theta}_2)$, and hence the posterior of $z_i$ collapses. Figure 1a illustrates a fit of this (non-identifiable) GMVAE to the pinwheel data [22]. In Section 3, we construct an latent-identifiable VAE (LIDVAE) that avoids this collapse.

Latent variable identifiability is a function of the both the model and the true data-generating distribution. Consider fitting the same GMVAE with $K = 2$ but to a different dataset of 5,000 samples, this one drawn from a GMVAE with only one cluster. (There is model misspecification.) One maximizing parameter value $\hat{\theta}_3$ is where both of the fitted clusters correspond to the true data generating cluster. While this parameter value resembles that of the first maximizer $\hat{\theta}_1$ above—both correspond to the true data generating cluster—this dataset leads to a different situation for latent variable identifiability. The two fitted clusters are the same and so different cluster memberships do not result in different likelihoods of $p(x_i|z_i;\hat{\theta}_3)$. The latent variable $z_i$ is not identifiable and its posterior collapses.

**Takeaways.** The GMVAE example in this section (and the PPCA and GMM examples in Appendix A) illustrate different ways that a latent variable can be non-identifiable in a model and suffer from posterior collapse. They show that even the true posterior—without variational inference—can collapse in non-identifiable models. They also illustrate that whether a latent variable is identifiable can depend on both the model and the data. Posterior collapse is an intrinsic problem of the model and the data, rather than specific to the use of neural networks or variational inference.

The equivalence between posterior collapse and latent variable non-identifiability in Theorem 1 also implies that, to mitigate posterior collapse, we should try to resolve latent variable non-identifiability. In the next section, we develop such a class of latent-identifiable VAE.

## 3 Latent-identifiable VAE via Brenier maps

We now construct latent-identifiable VAE, a class of VAE whose latent variables are guaranteed to be identifiable, and thus the posteriors cannot collapse.

### 3.1 The latent-identifiable VAE

To construct the latent-identifiable VAE, we rely on a key observation that, to guarantee latent variable identifiability, it is sufficient to make the likelihood function $P(x_i|z_i;\theta)$ injective for all values of $\theta$. If the likelihood is injective, then, for any $\theta$, each value of $z_i$ will lead to a different distribution $P(x_i|z_i;\theta)$. In particular, this fact will be true for any optimized $\hat{\theta}$ and so the latent $z_i$ must be identifiable, regardless of the data. By Theorem 1, its posterior cannot collapse.

Constructing latent-identifiable VAE thus amounts to constructing an injective likelihood function for VAE. The construction is based on a few building blocks of linear and nonlinear injective functions, then composed into an injective likelihood $p(x_i|z_i;\theta)$ mapping from $\mathcal{Z}^d$ to $\mathcal{X}^m$, where $\mathcal{Z}$ and $\mathcal{X}$ indicate the set of values $z_i$ and $x_i$ can take. For example, if $x_i$ is an m-dimensional binary vector, then $\mathcal{X} = \{0,1\}^m$; if $z_i$ is a $K$-dimensional real-valued vector, then $\mathcal{Z} = \mathbb{R}^d$.

**The building blocks of LIDVAE: Injective functions.** For linear mappings from $\mathbb{R}^{d_1}$ to $\mathbb{R}^{d_2}$ ($d_2 \geq d_1$), we consider matrix multiplication by a $d_1 \times d_2$-dimensional matrix $\beta$. For a $d_1$-dimensional variable $z$, left multiplication by a matrix $\beta^\top$ is injective when $\beta$ has full column rank [53]. For example, a matrix with all ones in the diagonal and all other entries being zero has full column rank.

For nonlinear injective functions, we focus on Brenier maps [4, 37]. A $d$-dimensional Brenier map is is the gradient of a convex function from $\mathbb{R}^d$ to $\mathbb{R}$. That is, a Brenier map satisfies $g = \nabla T$ for some

convex function $T : \mathbb{R}^d \to \mathbb{R}$. Brenier maps are also known as a monotone transport map. They are guaranteed to be bijective [4, 37] because their derivative is the Hessian of a convex $T$, which must be positive semidefinite and has a nonnegative determinant [4].

To build a VAE with Brenier maps, we require a neural network parametrization of the Brenier map. As Brenier maps are gradients of convex functions, we begin with the neural network parametrizaton of convex functions, namely the input convex neural network (ICNN) [2, 35]. This parameterization of convex functions will enable Brenier maps to be paramterized as the gradient of ICNN.

An $L$-layer ICNN is a neural network mapping from $\mathbb{R}^d$ to $\mathbb{R}$. Given an input $u \in \mathbb{R}^d$, its $l$th layer is

$$\boldsymbol{z}_0 = \mathbf{u}, \qquad \boldsymbol{z}_{l+1} = h_l(\mathbf{W}_l \boldsymbol{z}_l + \mathbf{A}_l \mathbf{u} + \mathbf{b}_l), \qquad (l = 0, \dots, L-1), \tag{6}$$

where the last layer $\boldsymbol{z}_L$ must be a scalar, $\{\mathbf{W}_l\}$ are non-negative weight matrices with $\mathbf{W}_0 = \mathbf{0}$. The functions $\{h_l : \mathbb{R} \to \mathbb{R}\}$ are convex and non-decreasing entry-wise activation functions for layer $l$; they are applied element-wise to the vector $(\mathbf{W}_l \boldsymbol{z}_l + \mathbf{A}_l \mathbf{u} + \mathbf{b}_l)$. A common choice of $h_0 : \mathbb{R} \to \mathbb{R}$ is the square of a leaky RELU, $h_0(x) = (\max(\alpha \cdot x, x))^2$ with $\alpha = 0.2$; the remaining $h_l$'s are set to be a leaky RELU, $h_l(x) = \max(\alpha \cdot x, x)$. This neural network is called "input convex" because it is guaranteed to be a convex function.

Input convex neural networks can approximate any convex function on a compact domain in sup norm (Theorem 1 of Chen et al. [9].) Given the neural network parameterization of convex functions, we can parametrize the Brenier map $g_\theta(\cdot)$ as its gradient with respect to the input $g_\theta(u) = \partial z_L / \partial u$. This neural network parameterization of Brenier map is a universal approxiamtor of all Brenier maps on a compact domain, because input convex neural networks are universal approximators of convex functions [9].

**The latent-identifiable VAE (LIDVAE).** We construct injective likelihoods for LIDVAE by composing two bijective Brenier maps with an injective matrix multiplication. As the composition of injective and bijective mappings must be injective, the resulting composition must be injective. Suppose $g_{1,\theta} : \mathbb{R}^K \to \mathbb{R}^K$ and $g_{2,\theta} : \mathbb{R}^D \to \mathbb{R}^D$ are two Brenier maps, and $\beta$ is a $K \times D$-dimensional matrix ($D \geq K$) with all the main diagonal entries being one and all other entries being zero. The matrix $\beta^\top$ has full column rank, so multiplication by $\beta^\top$ is injective. Thus the composition $g_{2,\theta}(\beta^\top g_{1,\theta}(\cdot))$ must be an injective function from a low-dimensional space $\mathbb{R}^K$ to a high-dimensional space $\mathbb{R}^D$.

**Definition 3** (Latent-identifiable VAE (LIDVAE) via Brenier maps)**.** *An* LIDVAE *via Brenier maps generates a $D$-dimensional datapoint $x_i, \in \{1, \dots, n\}$ by:*

$$z_i \sim p(z_i), \qquad x_i \mid z_i \sim \mathrm{EF}(x_i \mid g_{2,\theta}(\beta^\top g_{1,\theta}(z_i))), \tag{7}$$

*where* EF *stands for exponential family distributions; $z_i$ is a $K$-dimensional latent variable, discrete or continuous. The parameters of the model are $\theta = (g_{1,\theta}, g_{2,\theta})$, where $g_{1,\theta} : \mathbb{R}^K \to \mathbb{R}^K$ and $g_{2,\theta} : \mathbb{R}^D \to \mathbb{R}^D$ are two continuous Brenier maps. The matrix $\beta$ is a $K \times D$-dimensional matrix ($D \geq K$) with all the main diagonal entries being one and all other entries being zero.*

Contrasting LIDVAE (Eq. 7) with the classical VAE (Eq. 1), the LIDVAE replaces the function $f_\theta : \mathcal{Z}^K \to \mathcal{X}^D$ with the injective mapping $g_{2,\theta}(\beta^\top g_{1,\theta}(\cdot))$, composed by bijective Brenier maps $g_{1,\theta}, g_{2,\theta}$ and a zero-one matrix $\beta^\top$ with full column rank. As the likelihood functions of exponential family are injective, the likelihood function $p(x_i \mid z_i; \theta) = \mathrm{EF}(g_{2,\theta}(\beta^\top g_{1,\theta}(z_i)))$ of LIDVAE must be injective. Therefore, replacing an arbitrary function $f_\theta : \mathcal{Z}^K \to \mathcal{X}^D$ with the injective mapping $g_{2,\theta}(\beta^\top g_{1,\theta}(\cdot))$ plays a crucial role in enforcing identifiability for latent variable $z_i$ and avoiding posterior collapse in LIDVAE. As the latent $z_i$ must be identifiable in LIDVAE, its posterior does not collapse.

Despite its injective likelihood, LIDVAE are as flexible as VAE; the use of Brenier maps and ICNN does not limit the capacity of the generative model. Loosely, LIDVAE can model any distributions in $\mathbb{R}^D$ because Brenier maps can map any given non-atomic distribution in $\mathbb{R}^d$ to any other one in $\mathbb{R}^d$ [37]. Moreover, the ICNN parametrization is a universal approximator of Brenier maps [2]. We summarize the key properties of LIDVAE in the following proposition.

**Proposition 2.** *The latent variable $z_i$ is identifiable in* LIDVAE*, i.e. for all $i \in \{1, \dots, n\}$, we have*

$$p(x_i \mid z_i = \tilde{z}'; \theta) = p(x_i \mid z_i = \tilde{z}; \theta) \qquad \Rightarrow \qquad \tilde{z}' = \tilde{z}, \qquad \forall \tilde{z}', \tilde{z}, \theta. \tag{8}$$

*Moreover, for any* VAE*-generated data distribution, there exists an* LIDVAE *that can generate the same distribution. (The proof is in Appendix B.)*

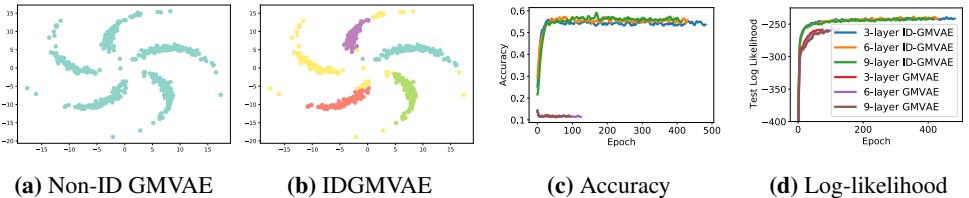

| (a) Non-ID GMVAE | (b) IDGMVAE | (c) Accuracy | (d) Log-likelihood |

**Figure 1:** (a)-(b): The posterior of the classical GMVAE [13, 26, 51] collapses when fit to the pinwheel dataset; the latents predict the same value for all datapoints. The posteriors of latent-identifiable Gaussian mixture VAE (LIDGMVAE), however, do not collapse and provide meaningful representations. (c)-(d) The latent-identifiable GMVAE produces posteriors that are substantially more informative than GMVAE when fit to fashion MNIST. It also achieves higher test log likelihood.

## 3.2   Inference in LIDVAE

Performing inference in LIDVAE is identical to the classical VAE, as the two VAE differ only in their parameter constraints. To fit an LIDVAE, we use the classical amortized inference algorithm of VAE; we maximize the evidence lower bound (ELBO) of the log marginal likelihood [28].

In general, LIDVAE are a drop-in replacement for VAE. Both have the same capacity (Proposition 2) and share the same inference algorithm, but LIDVAE is identifiable and does not suffer from posterior collapse. The price we pay for LIDVAE is computational: the generative model (i.e. decoder) is parametrized using the gradient of a neural network; its optimization thus requires calculating gradients of the gradient of a neural network, which increases the computational complexity of VAE inference and can sometimes challenge optimization. While fitting classical VAE using stochastic gradient descent has $O(k \cdot p)$ computational complexity, where $k$ is the number of iterations and $p$ is the number of parameters, fitting latent-identifiable VAE may require $O(k \cdot p^2)$ computational complexity.

## 3.3   Extensions of LIDVAE

The construction of LIDVAE reveals a general strategy to make the latent variables of generative models identifiable: replacing nonlinear mappings with injective nonlinear mappings. We can employ this strategy to make the latent variables of many other VAE variants identifiable. Below we give two examples, mixture VAE and sequential VAE.

The mixture VAE, with GMVAE as a special case, models the data with an exponential family mixture and mapped through a flexible neural network to generate the data. We develop its latent-identifiable counterpart using Brenier maps.

**Example 1** (Latent-identifiable mixture VAE (LIDMVAE))**.** *An* LIDMVAE *generates a D-dimensional datapoint* $x_i, i \in \{1, \dots, n\}$ *by*

$$z_i \sim \text{Categorical}(1/K), \quad w_i \,|\, z_i \sim \text{EF}(w_i \,|\, \beta_1^\top z_i), \quad x_i \,|\, w_i \sim \text{EF}(x_i \,|\, g_{2,\theta}(\beta_2^\top g_{1,\theta}(w_i))), \quad (9)$$

*where* $W_i$ *is a K-dimensional one-hot vector that indicates the cluster assignment. The parameters of the model are* $\theta = (g_{1,\theta}, g_{2,\theta})$, *where the functions* $g_{1,\theta} : \mathbb{R}^M \to \mathbb{R}^M$ *and* $g_{2,\theta} : \mathbb{R}^D \to \mathbb{R}^D$ *are two continuous Brenier maps. The matrices* $\beta_1$ *and* $\beta_2$ *are a* $K \times M$-*dimensional matrix* $(M \geq K)$ *and a* $M \times D$-*dimensional matrix* $(D \geq M)$ *respectively, both having all the main diagonal entries being one and all other entries being zero.*

The LIDMVAE differs from the classical mixture VAE in $p(x_i \,|\, z_i)$, where we replace its neural network mapping with its injective counterpart, i.e. a composition of two Brenier maps and a matrix multiplication $g_{2,\theta}(\beta_2^\top g_{1,\theta}(\cdot))$. As a special case, setting both exponential families in Example 1 as Gaussian gives us LIDGMVAE, which we will use to model images in Section 4.

Next we derive the identifiable counterpart of sequential VAE, which models the data with an autoregressive model conditional on the latents.

**Example 2** (Latent-identifiable sequential VAE (LIDSVAE))**.** *An* LIDSVAE *generates a D-dimensional datapoint* $x_i, i \in \{1, \dots, n\}$ *by*

$$z_i \sim p(z_i), \quad x_i \,|\, z_i, x_{<i} \sim \text{EF}(g_{2,\theta}(\beta_2^\top g_{1,\theta}([z_i, f_\theta(x_{<i})]))),$$

| | Fashion-MNIST | | | | Omniglot | | | |
|---|---|---|---|---|---|---|---|---|
| | **AU** | **KL** | **MI** | **LL** | **AU** | **KL** | **MI** | **LL** |
| VAE [28] | 0.1 | 0.2 | 0.9 | -258.8 | 0.02 | 0.0 | 0.1 | -862.1 |
| SA-VAE [25] | 0.2 | 0.3 | 1.3 | -252.2 | 0.1 | 0.2 | 1.0 | -853.4 |
| Lagging VAE [18] | 0.4 | 0.6 | 1.6 | -248.5 | 0.5 | 1.0 | 3.6 | -849.4 |
| $\beta$-VAE [19] ($\beta$=0.2) | 0.6 | 1.2 | 2.4 | -245.3 | 0.7 | 1.4 | 5.9 | -842.6 |
| **LIDGMVAE (this work)** | **1.0** | **1.6** | **2.6** | **-242.3** | **1.0** | **1.7** | **7.5** | **-820.3** |

| | Synthetic | | | | Yahoo | | | | Yelp | | | |
|---|---|---|---|---|---|---|---|---|---|---|---|---|
| | **AU** | **KL** | **MI** | **LL** | **AU** | **KL** | **MI** | **LL** | **AU** | **KL** | **MI** | **LL** |
| VAE [28] | 0.0 | 0.0 | 0.0 | -46.5 | 0.0 | 0.0 | 0.0 | -519.7 | 0.0 | 0.0 | 0.0 | -635.9 |
| SA-VAE [25] | 0.4 | 0.1 | 0.1 | -40.2 | 0.2 | 1.0 | 0.2 | -520.2 | 0.1 | 1.9 | 0.2 | -631.5 |
| Lagging VAE [18] | 0.5 | 0.1 | 0.1 | -40.0 | 0.3 | 1.6 | 0.4 | **-518.6** | 0.2 | 3.6 | 0.1 | **-631.0** |
| $\beta$-VAE [19] ($\beta$=0.2) | **1.0** | 0.1 | 0.1 | **-39.9** | 0.5 | 4.7 | 0.9 | -524.4 | 0.3 | **10.0** | 0.1 | -637.3 |
| LIDSVAE | **1.0** | **0.5** | **0.6** | -40.3 | **0.8** | **7.2** | **1.1** | -519.5 | **0.7** | 9.1 | **0.9** | -634.2 |

**Table 1:** Across image and text datasets, LIDVAE outperforms existing VAE variants in preventing posterior collapse while achieving similar goodness-of-fit to the data.

where $x_{<i} = (x_1, \ldots, x_{i-1})$ represents the history of $x$ before the $i$th dimension. The function $f_\theta : \mathcal{X}_{<i} \to \mathbb{R}^H$ maps the history $X_{<i}$ into an $H$-dimensional vector. Finally, $[z_i, f_\theta(x_{<i})]$ is an $(K+H) \times 1$ vector that represents a row-stack of the vectors $(z_i)_{K \times 1}$ and $(f_\theta(x_{<i}))_{H \times 1}$.

Similar with mixture VAE, the LIDSVAE also differs from sequential VAE only in its use of $g_{2,\theta}(\beta_2^\top g_{1,\theta}(\cdot))$ function in $p(x_i | z_i, x_{<i})$. We will use LIDSVAE to model text in Section 4.

## 4   Empirical studies

We study LIDVAE on images and text datasets, finding that LIDVAE do not suffer from posterior collapse as we increase the capacity of the generative model, while achieving similar fits to the data. We further study PPCA, showing how likelihood functions nearly constant in latent variables lead to collapsing posterior even with Markov chain Monte Carlo (MCMC).

### 4.1   LIDVAE on images and text

We consider three metrics for evaluating posterior collapse: (1) KL divergence between the posterior and the prior, $\mathrm{KL}(q(\boldsymbol{z}|\boldsymbol{x})||p(\boldsymbol{z}))$; (2) Percentage of active units (AU):AU $= \sum_{d=1}^{D} \mathbb{1}\{\mathrm{Cov}_{p(\boldsymbol{x})}(\mathbb{E}_{q(\boldsymbol{z}|\boldsymbol{x})}[\boldsymbol{z}_d]) \geq \epsilon\}$, where $\boldsymbol{z}_d = (z_{1d}, \ldots, z_{nd})$ is the $d$th dimension of the latent variable $\boldsymbol{z}$ for all the $n$ data points. In calculating AU, we follow Burda et al. [7] to calculate the posterior mean, $(\mathbb{E}[z_{1d}|\boldsymbol{x}_1], \ldots, \mathbb{E}[z_{nd}|\boldsymbol{x}_n])$ for all data points, and calculate the sample variance of $\mathbb{E}[z_{id}|\boldsymbol{x}_i]$ across $i$'s from this vector. The threshold $\epsilon$ is chosen to be 0.01 [7]; the theoretical maximum of %AU is one; (3) Approximate Mutual information (MI) between $\boldsymbol{x}_i$ and $\boldsymbol{z}_i$, $I(\boldsymbol{x}, \boldsymbol{z}) = \mathbb{E}_{\boldsymbol{x}}\left[\mathbb{E}_{q(\boldsymbol{z}|\boldsymbol{x})}[\log(q(\boldsymbol{z}|\boldsymbol{x}))]\right] - \mathbb{E}_{\boldsymbol{x}}\left[\mathbb{E}_{q(\boldsymbol{z}|\boldsymbol{x})}[\log(q(\boldsymbol{z}))]\right]$. We also evaluate the model fit using the importance weighted estimate of log-likelihood on a held-out test set [7]. For mixture VAE, we also evaluate the predictive accuracy of the categorical latents against ground truth labels to quantify their informativeness.

**Competing methods.** We compare LIDVAE with the classical VAE [28], the $\beta$-VAE ($\beta$=0.2) [19], the semi-amortized VAE [25], and the lagging VAE [18]. Throughout the empirical studies, we use flexible variational approximating families (RealNVPs [14] for image and LSTMs [20] for text).

**Results: Images.** We first study LIDGMVAE on four subsampled image datasets drawn from pinwheel [22], MNIST [31], Fashion MNIST [57], and Omniglot [30]. Figures 1a and 1b illustrate a fit of the GMVAE and the LIDGMVAE to the pinwheel data [22]. The posterior of the GMVAE latents collapse, attributing all datapoints to the same latent cluster. In contrast, LIDGMVAE produces categorical latents faithful to the clustering structure. Figure 1 examines the LIDGMVAE as we increase the flexibility of the generative model. Figure 1c shows that the categorical latents of the LIDGMVAE are substantially more predictive of the true labels than their classical counterparts. Moreover, its performance does not degrade as the generative model becomes more flexible. Figure 1d

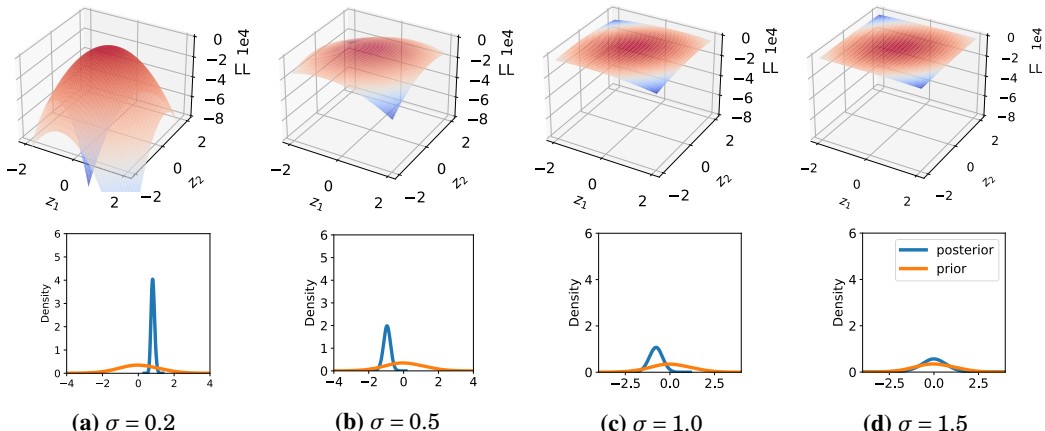

**Figure 2:** As the noise level increases in PPCA, the latent variable becomes closer to non-identifiable because the likelihood and more susceptible to posterior collapse. Its likelihood surface becomes flatter and its posterior becomes closer to the prior. Top panel: Likelihood surface of PPCA as a function of the two latents $z_1, z_2$. When $\sigma$ increase, the likelihood surface becomes flatter and the latent variables $z_1, z_2$ are closer to non-identifiable. Bottom panel: Posterior of $z_1$ under different $\sigma$ values. When $\sigma$ increase, the posterior becomes closer to the prior.

shows that the LIDGMVAE consistently achieve higher test log-likelihood. Table 1 compares different variants of VAE in a 9-layer generative model. Across four datasets, LIDGMVAE mitigates posterior collapse. It achieves higher AU, KL and MI than other variants of VAE. It also achieves a higher test log-likelihood.

**Results: Text.** We apply LIDSVAE to three subsampled text datasets drawn from a synthetic text dataset, the Yahoo dataset, and the Yelp dataset [60]. The synthetic dataset is generated from a classical two-layer sequential VAE with a five-dimensional latent. Table 1 compares the LIDSVAE with the sequential VAE. Across the three text datasets, the LIDSVAE outperforms other variants of VAE in mitigating posterior collapse, generally achieving a higher AU, KL, and MI.

## 4.2 Latent variable non-identifiability and posterior collapse in PPCA

Here we show that the PPCA posterior becomes close to the prior when the latent variable becomes close to be non-identifiable. We perform inference using Hamiltonian Monte Carlo (HMC), avoiding the effect of variational approximation on posterior collapse.

Consider a PPCA with two latent dimensions, $p(z_i) = \mathcal{N}(z_i; 0, I_2)$, $p(x_i | z_i; \theta) = \mathcal{N}(x_i; z_i^\top w, \sigma^2 \cdot I_5)$, where the value of $\sigma^2$ is known, $z_i$'s are the latent variables of interest, and $w$ is the only parameter of interest. When the noise $\sigma^2$ is set to a large value, the latent variable $z_i$ may become nearly non-identifiable. The reason is that the likelihood function $p(x_i | z_i)$ becomes slower-varying as $\sigma^2$ increases. For example, Figure 2 shows that the likelihood surface becomes flatter as $\sigma^2$ increases. Accordingly, the posterior becomes closer to the prior as $\sigma^2$ increases. When $\sigma = 1.5$, the posterior collapses. This non-identifiability argument provides an explanation to the closely related phenomenon described in Section 6.2 of [33].

## 5 Discussion

In this work, we show that the posterior collapse phenomenon is a problem of latent variable non-identifiability. It is not specific to the use of neural networks or particular inference algorithms in VAE. Rather, it is an intrinsic issue of the model and the dataset. To this end, we propose a class of LIDVAE via Brenier maps to resolve latent variable non-identifiability and mitigate posterior collapse. Across empirical studies, we find that LIDVAE outperforms existing methods in mitigating posterior collapse.

The latent variables of LIDVAE are guaranteed to be identifiable. However, it does not guarantee that the latent variables and the parameters of LIDVAE are jointly identifiable. In other words, the LIDVAE model may not be identifiable even though its latents are identifiable. This difference between latent variable identifiability and model identifiability may appear minor. But the tractability of resolving latent variable identifiability plays a key role in making non-identifiability a fruitful one perspective of posterior collapse. To enforce latent variable identifiability, it is sufficient to ensure that the likelihood $p(\boldsymbol{x}|\boldsymbol{z}, \hat{\theta})$ is an injective function of $\boldsymbol{z}$. In contrast, resolving model identifiability for the general class of VAE remains a long standing open problem, with some recent progress relying on auxiliary variables [23, 24]. The tractability of resolving latent variable identifiability is a key catalyst of a principled solution to mitigating posterior collapse.

There are a few limitations of this work. One is that the theoretical argument focuses on the collapse of the exact posterior. The rationale is that, if the exact posterior collapses, then its variational approximation must also collapse because variational approximation of posteriors cannot "uncollapse" a posterior. That said, variational approximation may "collapse" a posterior, i.e. the exact posterior does not collapse but the variational approximate posterior collapses. The theoretical argument and algorithmic approaches developed in this work does not apply to this setting, which remains an interesting venue of future work.

A second limitation is that the latent-identifiable VAE developed in this work bear a higher computational cost than classical VAE. While the latent-identifiable VAE ensures the identifiability of its latent variables and mitigates posterior collapse, it does come with a price in computation because its generative model (i.e. decoder) is parametrized using gradients of a neural network. Fitting the latent-identifiable VAE thus requires calculating gradients of gradients of a neural network, leading to much higher computational complexity than fitting the classical VAE. Developing computationally efficient variants of the latent-identifiable VAE is another interesting direction for future work.

**Acknowledgments.** We thank Taiga Abe and Gemma Moran for helpful discussions, and anonymous reviewers for constructive feedback that improved the manuscript. David Blei is supported by ONR N00014-17-1-2131, ONR N00014-15-1-2209, NSF CCF-1740833, DARPA SD2 FA8750-18-C-0130, Amazon, and the Simons Foundation. John Cunningham is supported by the Simons Foundation, McKnight Foundation, Zuckerman Institute, Grossman Center, and Gatsby Charitable Trust.

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
