# Supplementary Materials

# Posterior Collapse and Latent Variable Non-identifiability

## A    Examples of posterior collapse continued

We present two additional examples of posterior collapse, probabilistic principal component analysis and Gaussian mixture model.

### A.1    Probabilistic principal component analysis

We consider classical probabilistic principal component analysis (PPCA) and show that its local latent variables can suffer from posterior collapse at maximum likelihood parameter values (i.e. global maxima of log marginal likelihood). This example refines the perspective of Lucas et al. [7], which demonstrated that posterior collapse can occur in PPCA absent any variational approximation but due to local maxima in the log marginal likelihood. Here we show that posterior collapse can occur even with global maxima, absent optimization issues due to local maxima.

Consider a PPCA with two latent dimensions,

$$p(z_i) = \mathcal{N}(z_i \,|\, 0, I_2),$$
$$p(x_i \,|\, z_i \,; \theta) = \mathcal{N}(x_i \,|\, z_i^\top w, \sigma^2 \cdot I_5),$$

where $z_i$'s are the latent variables of interest and others $\theta = (w, \sigma^2)$ are parameters of the model.

Consider fitting this model to two datasets, each with 500 samples, focusing on maximum likelihood parameter values. Depending on the true distribution of the dataset, PPCA may or may not suffer from posterior collapse.

1. Sample the data from a one-dimensional PPCA,

$$x_i \sim \mathcal{N}(x_i \,|\, \mathcal{N}(0, I_1) \cdot \bar{w}_1, \bar{\sigma}_1 \cdot I_5). \tag{10}$$

   (The model remains two dimensional.) The latent variables $z_i$'s are not (fully) identifiable in this case. The reason is that one set of maximum likelihood parameters is $\hat{\theta} = (\hat{w}, \hat{\sigma}) = ([\mathbf{0}, \bar{w}_1], \bar{\sigma}_1)$, i.e. setting one latent dimension as zero and the other equal to the true data generating direction. Under this $\hat{\theta}$, the likelihood function is constant in the first dimension of the latent variable, i.e. $z_{i1}$; see Figure 3a. The posterior of $z_{i1}$ thus collapses, matching the prior, while the posterior of $z_{i2}$ stays peaked (Figure 3b).

2. Sample the data from from a two-dimensional PPCA,

$$x_i \sim \mathcal{N}(x_i \,|\, \mathcal{N}(0, I_2) \cdot \bar{w}_2, \bar{\sigma}_2 \cdot I_5). \tag{11}$$

   The latent variables $z_i$ are identifiable. The likelihood function varies against both $z_{i1}$ and $z_{i2}$; the posteriors of both $z_{i1}$ and $z_{i2}$ are peaked (Figures 3c and 3d).

### A.2    Gaussian mixture model

Though we have focused on the posterior collapse of local latent variables, a model can also suffer from posterior collapse of its global latent variables. Consider a simple Gaussian mixture model (GMM) with two clusters,

$$p(\alpha) = \mathrm{Beta}(\alpha \,|\, 5, 5),$$
$$p(x_i \,|\, \alpha \,; \theta) = \alpha \cdot \mathcal{N}(x_i \,|\, \mu_1, \sigma_1^2) + (1 - \alpha) \cdot \mathcal{N}(x_i \,|\, \mu_2, \sigma_2^2).$$

Here $\alpha$ is a global latent variable and $\theta = (\mu_1, \mu_2, \sigma_1, \sigma_2)$ are the parameters of the model. Fit this model to three datasets, each with $10^5$ samples.

1. Sample the data from two non-overlapping clusters,

$$x_i \sim 0.15 \cdot \mathcal{N}(-10, 1) + 0.85 \cdot \mathcal{N}(10, 1). \tag{12}$$

   The latent variable $\alpha$ is identifiable. The two data generating clusters are substantially different, so the likelihood function varies across $\alpha \in [0, 1]$ under the maximum likelihood (ML) parameters (Figure 4a). The posterior of $\alpha$ is also peaked (Figure 4b) and differs much from the prior.

2. Sample the data from two overlapping clusters,

$$x_i \sim 0.15 \cdot \mathcal{N}(-0.5, 1) + 0.85 \cdot \mathcal{N}(0.5, 1). \tag{13}$$

The latent variable $\alpha$ is identifiable. However, it is nearly non-identifiable. While the two data generating clusters are different, they are very similar to each other because they overlap. Therefore, the likelihood function $p(x_i | \alpha; \theta^*)$ is slowly varying under ML parameters $\theta^* = (\mu_1^*, \mu_2^*, \sigma_1^*, \sigma_2^*) = (-0.5, 0.5, 1, 1)$; see Figure 4a. Consequently, the posterior of $\alpha$ remains very close to the prior; see Figure 4b.

3. Sample the data from a single Gaussian distribution, $x_i \sim \mathcal{N}(-1, 1)$. The latent variable $\alpha$ is non-identifiable. The reason is that one set of ML parameters is $\theta^* = (\mu_1^*, \mu_2^*, \sigma_1^*, \sigma_2^*) = (-1, -1, 1, 1)$, i.e. setting both of the two mixture components equal to the true data generating Gaussian distribution. Under this $\theta^*$, the latent variable $\alpha$ is non-identifiable and its likelihood function $p(\{x_i\}_{i=1}^n | \alpha; \theta^*)$ is constant in $\alpha$ because the two mixture components are equal; Figure 4a illustrates this fact. Moreover, the posterior of $\alpha$ collapses, $p(\alpha | \{x_i\}_{i=1}^n; \theta^*) = p(\alpha)$. Figure 4b illustrates this fact: The HMC samples of the $\alpha$ posterior closely match those drawn from the prior. (Exact inference is intractable in this case, so we use HMC as a close approximation to exact inference.) This example demonstrates the connection between non-identifiability and posterior collapse; it also shows that posterior collapse is not specific to variational inference but is an issue of the model and the data.

As for PPCA, these GMM examples demonstrate that whether a latent variable is identifiable in a probabilistic model not only depends on the model but also the data. While all three examples were fitted with the same GMM model, their identifiability situation differs as the samples are generated in different ways.

## B Proof of Proposition 2

We prove a general version of Proposition 2 by establishing the latent variable identifiability and flexibility of the most general form of the LIDVAE. The LIDVAE, LIDMVAE, and LIDSVAE (Definition 4 and examples 1 and 2) will all be its special cases. Then Proposition 2 will also be a special case of the more general result stated below (Proposition 3).

We first define the most general form of LIDVAE.

**Definition 4** (General LIDVAE via Brenier maps). *A general* LIDVAE *via Brenier maps generates an D-dimensional data-point $x_i, \in \{1, \ldots, n\}$ by:*

$$(z_i)_{K \times 1} \sim p(z_i), \tag{14}$$

$$(w_i)_{M \times 1} | z_i \sim \mathrm{EF}(w_i | \beta_1^\top z_i), \tag{15}$$

$$(x_i)_{D \times 1} | w_i, x_{<i} \sim \mathrm{EF}(x_i | h \circ g_{2,\theta}(\beta_2^\top g_{1,\theta}([w_i, f_\theta(x_{<i})]))), \tag{16}$$

*where* EF *stands for exponential family distributions; $z_i$ is a K-dimensional latent variable, discrete or continuous. The parameters of the model are $\theta = (g_{1,\theta}, g_{2,\theta}, f_\theta)$, where $f_\theta : \mathcal{X}_{<i} \to \mathbb{R}^H$ is a function that maps all previous data points $x_{<i}$ to an H-dimensional vector, $g_{1,\theta} : \mathbb{R}^{M+H} \to \mathbb{R}^{M+H}$ and $g_{2,\theta} : \mathbb{R}^D \to \mathbb{R}^D$ are two continuous monotone transport maps. The function $h(\cdot)$ is a bijective link function for the exponential family, e.g. the sigmoid function. The matrix $\beta_1$ is a $K \times M$-dimensional matrix ($M \geq K$) all the main diagonal entries being one and all other entries being zero, and thus with full row rank. Similarly, $\beta_2$ is a $(M+H) \times D$-dimensional matrix ($D \geq M+H$) with all the main diagonal entries being one and all other entries being zero, also with full row rank. Finally, $[w_i, f_\theta(x_{<i})]$ is an $(M+H) \times 1$ vector that represents a row-stack of the vectors $(w_i)_{M \times 1}$ and $(f_\theta(x_{<i}))_{H \times 1}$.*

The general LIDVAE differs from the classical VAE whose general form is

$$(z_i)_{K \times 1} \sim p(z_i), \tag{17}$$

$$(w_i)_{M \times 1} | z_i \sim \mathrm{EF}(w_i | \beta_1^\top z_i), \tag{18}$$

$$(x_i)_{D \times 1} | w_i, x_{<i} \sim \mathrm{EF}(x_i | h \circ g_\theta([w_i, f_\theta(x_{<i})])), \tag{19}$$

The key difference is in Eq. 19, where the classical VAE uses an arbitrary function $g : \mathbb{R}^{M+H} \to \mathbb{R}^D$ in Eq. 19. In contrast, LIDVAE uses a composition $g_{2,\theta}(\beta_2^\top g_{1,\theta}(\cdot))$ with additional constraints in Eq. 16.

General LIDVAE can handle both i.i.d. and sequential data. For i.i.d data (e.g. images), we can set $f_\theta(\cdot)$ to be a zero function, which implies $P(x_i | w_i, x_{<i}) = P(x_i | w_i)$. For sequential data (e.g. text), we can set $f_\theta(\cdot)$ to be an LSTM that embeds the history $x_{<i}$ into an $H$-dimensional vector.

General LIDVAE emulate many existing VAE. Letting $z_i$ be categorical (one-hot) vectors, the distribution $\mathrm{EF}(z_i^\top \beta_\theta)$ is an exponential family mixture. The identifiable VAE then maps this mixture model through a flexible function $g_\theta$. When $z_i$ is real-valued, it mimics classical VAE by mapping an exponential family PCA through flexible functions.

LIDGMVAE is a special case of the general LIDVAE when we set $z_i$ be categorical (one-hot) vectors, set the exponential family distribution EF to be Gaussian in Eqs. 15 and 16. In this case, $w_i \sim \mathrm{Gaussian}(z_i^\top \beta_\theta, \gamma_\theta)$ is a Gaussian mixture. Then, we set $f_\theta(\cdot)$ to be a zero function, which implies $P(x_i | w_i, x_{<i}) = P(x_i | w_i)$, and finally set $h$ as the identity function.

This general LIDVAE also subsumes the Bernoulli mixture model, which is a common variant of LIDGMVAE for the MNIST data. Specifically, we can set $z_i$ be categorical (one-hot) vectors, and then set the exponential family distribution EF to be Gaussian in Eq. 15, making $w_i \sim \mathrm{Gaussian}(z_i^\top \beta_\theta, \gamma_\theta)$ to be a Gaussian mixture. Next we set $f_\theta(\cdot)$ to be a zero function, which implies $P(x_i | w_i, x_{<i}) = P(x_i | w_i)$, then set $h$ to be the sigmoid function, and finally set the EF to be Bernoulli in Eq. 16.

LIDSVAE is another special case of the general LIDVAE when we set the EF to be a point mass and $\beta_{1,\theta}$ to be identity matrix in Eq. 15, which implies $w_i = z_i$. Then setting the EF to be a categorical distribution and $h$ to be identity in Eq. 16 leads to a configuration that is the same as Example 2.

LIDVAE can be made deeper with more layers by introducing additional full row-rank matrices $\beta_k$ (e.g. ones with all the main diagonal entries being one and all other entries being zero) and additional Brenier maps $g_{k,\theta}$. For example, we can expand Eq. 16 with an additional layer by setting

$$(x_i)_{D \times 1} | w_i, x_{<i} \sim \mathrm{EF}(g_{3,\theta}(\beta_3^\top g_{2,\theta}(\beta_2^\top g_{1,\theta}([w_i, f_\theta(x_{<i})])))).$$

Next we establish the latent variable identifiability and flexibility of this general class of LIDVAE, which will imply the identifiability and flexibility of all the special cases above.

**Proposition 3.** *The latent variable $z_i$ is identifiable in* LIDVAE*, i.e. for all $i \in \{1, \ldots, n\}$, we have*

$$p(x_i | z_i = \tilde{z}', x_{<i}; \theta) = p(x_i | z_i = \tilde{z}, x_{<i}; \theta) \quad \Rightarrow \quad \tilde{z}' = \tilde{z}, \qquad \forall \tilde{z}', \tilde{z}, \theta. \tag{20}$$

*Moreover, for any data distribution generated by the classical* VAE *(Eqs. 17 to 19), there exists an* LIDVAE *that can generate the same distribution.*

*Proof.* We first establish the latent variable identifiability. To show that the latent variable $z_i$ is identifiable, it is sufficient to show that the mapping from $z_i$ to $p(x_i | z_i; \theta)$ is injective for all $\theta$. The injectivity holds because all the transformations $(\beta_1, \beta_2, g_{1,\theta}, g_{2,\theta})$ involved in the mapping is injective, and their composition must be injective: the linear transformations $(\beta_1, \beta_2)$ have full row rank and hence are injective; the nonlinear transformations $(g_{1,\theta}, g_{2,\theta})$ are monotone transport maps and are guaranteed to be bijective [1, 9]; finally, the exponential family likelihood is injective.

We next establish the flexibility of the LIDVAE, by proving that any VAE-generated $p(\boldsymbol{x})$ can be generated by an LIDVAE. The proof proceeds in two steps: (1) we show any VAE-generated $p(\boldsymbol{x})$ can be generated by a VAE with injective likelihood $p(x_i | z_i; \theta)$; (2) we show any $p(\boldsymbol{x})$ generated by an injective VAE can be generated by an LIDVAE.

To prove (1), suppose $\beta_1$ does not have full row rank and $g_\theta$ is not injective. Then there exists some $Z' \in \mathbb{R}^d$, $d < K$, and injective $\beta'_{1,\theta}, g'_\theta$ such that the new VAE can represent the same $p(\boldsymbol{x})$. The reason is that we can always turn an non-injective function into an injective one by considering its quotient space. In particular, we consider the quotient space with the equivalence relation between $z, z'$ defined as $p(x | z; \theta) = p(x | z'; \theta)\}$, which induces a bijection into $\mathbb{R}^d$. When $p(z')$ is no longer standard Gaussian, there must exist a bijective Brenier map $\tilde{z} = f_t(z')$ such that $p(\tilde{x})$ is standard Gaussian (Theorem 6 of McCann et al. [8]).

To prove (2), we show that any VAE with injective mapping can be reparameterized as a LIDVAE. To prove this claim, it is sufficient to show that any injective function $l_\theta : \mathbb{R}^{M+H} \to \mathbb{R}^D$ can be reparametrized as $g_{2,\theta}(\beta_2^\top g_{1,\theta}(\cdot))$. Below we provide such a reparametrization by solving for $g_1, g_2$

| | | Pinwheel | | | | MNIST | | |
|---|---|---|---|---|---|---|---|---|
| | **AU** | **KL** | **MI** | **LL** | **AU** | **KL** | **MI** | **LL** |
| VAE [6] | 0.2 | 1.4e-6 | 2.0e-3 | **-6.2 (5e-2)** | 0.1 | 0.1 | 0.2 | -108.2 (5e-1) |
| SA-VAE [5] | 0.2 | 1.6e-5 | 2.0e-2 | -6.5 (5e-2) | 0.4 | 0.4 | 0.6 | -106.3 (7e-1) |
| Lagging VAE [3] | 0.6 | 0.7e-3 | 1.5e0 | -6.5 (4e-2) | 0.5 | 0.8 | 1.7 | -105.2 (5e-1) |
| $\beta$-VAE [4] ($\beta$=0.2) | **1.0** | **1.2e-3** | **2.3e0** | -6.6 (6e-2) | 0.8 | 1.5 | 2.8 | -100.4 (6e-1) |
| LIDGMVAE (this work) | **1.0** | **1.2e-3** | 2.2e0 | -6.5 (5e-2) | **1.0** | **1.8** | **3.9** | **-95.4 (7e-1)** |

**Table 2:** LIDGMVAE do not suffer from posterior collapse and achieves better fit than its classical counterpart in a 9-layer generative model. The reported number is mean (sd) over ten different random seeds. (Higher is better.)

and $\beta$ in $l_\theta(z) = g_{2,\theta}(\beta_2^\top g_{1,\theta}(z))$. We set $g_{1,\theta}$ as an identity map, $\beta_2$ as an $(M+H) \times D$ matrix with all the main diagonal entries being one and all other entries being zero, and $g_{2,\theta}$ as an invertible $\mathbb{R}^d \to \mathbb{R}^d$ mapping which coincides with $l_\theta$ on the $(M+H)$-dimensional subspace of $z$.

Finally, we note that the same argument applies to the variant of VAE where $w_i = z_i$. It coincides with the classical VAE in Kingma & Welling [6]. Applying the same argument as above establishes Proposition 2.

$\square$

## C  Experiment details

For image experiments, all hidden layers of the neural networks have 512 units. We choose the number of continuous latent variables as 64 and the dimensionality of categorical variables as the number of ground truth labels. Then we use two-layer RealNVP ([2]) as an approximating family to tease out the effect of variational inference.

For text experiments, all hidden layers of the neural networks have 1024 units. We choose the dimensionality of the embedding as 1024. Then we use two-layer LSTM as an approximating family following common practice of fitting sequential VAE.

## D  Additional experimental results

Table 2 includes additional experimental results of LIDVAE on image datasets (Pinwheel and MNIST).

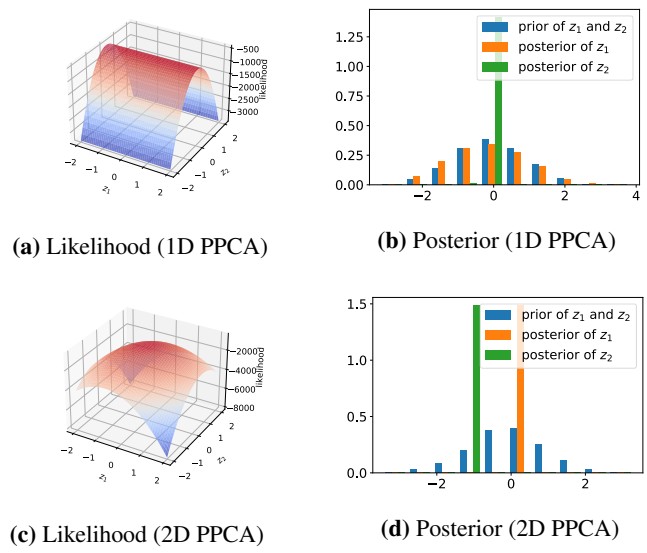

**(a)** Likelihood (1D PPCA)

**(b)** Posterior (1D PPCA)

**(c)** Likelihood (2D PPCA)

**(d)** Posterior (2D PPCA)

**Figure 3:** Fitting PPCA with more latent dimensions than enough leads to non-identifiable local latent variables and collapsed posteriors. (a)-(b) Fit a two-dimensional PPCA to data drawn from a one-dimensional PPCA. The likelihood surface is constant in one dimension of the latent variable, i.e. this latent variable is non-identifiable. Hence its corresponding posterior collapses. (c)-(d) Fit a two-dimensional PPCA to data from a two-dimensional PPCA does not suffer from posterior collapse; its likelihood surface varies in all dimensions.

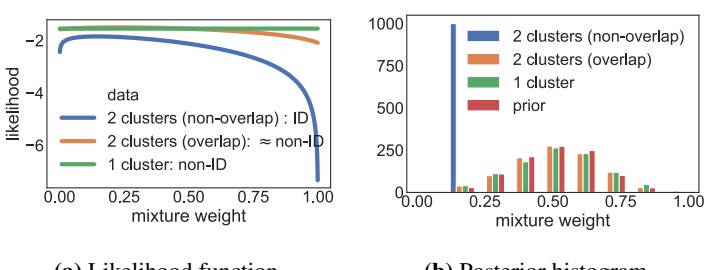

**(a)** Likelihood function

**(b)** Posterior histogram

**Figure 4:** When a latent variable is non-identifiable (non-ID) in a model, its likelihood function is a constant function and its posterior is equal to the prior, i.e. its posterior collapses. Consider a Gaussian mixture model with two clusters $x \sim \alpha \cdot \mathcal{N}(\mu_1, \sigma_1^2) + (1 - \alpha) \cdot \mathcal{N}(\mu_2, \sigma_2^2)$, treating the mixture weight $\alpha$ as the latent variable and others as parameters. Fit the model to datasets generated respectively by one Gaussian cluster ($\alpha$ non-identifiable), two overlapping Gaussian clusters ($\alpha$ nearly non-identifiable), and two non-overlapping Gaussian clusters ($\alpha$ identifiable). Under optimal parameters, the likelihood function $p(x|\alpha)$ is (close to) a constant when the latent variable $\alpha$ is (close to) non-identifiable; its posterior is also (close to) the prior. Otherwise, the likelihood function is non-constant and the posterior is peaked.