# OpenReview forum: "Posterior Collapse and Latent Variable Non-identifiability"
_NeurIPS.cc/2021/Conference — NeurIPS 2021 Poster_

### Official Review · Reviewer_dUTF · 2021-07-15

**Rating:** 7
**Confidence:** 3

**Summary:**

The paper proposes an explanation for the phenomenon of posterior collapse in variational autoencoders and proves that an exact notion of posterior collapse happens if and only if the latent variables are non-identifiable. Then, it proposes a new variational autoencoder model based on Brenier maps which ensures that the latent variables are identifiable. The new model can simply be “plugged into” existing variational autoencoder models. Experiments demonstrate that the new model mitigates posterior collapse.

**Limitations And Societal Impact:**

I mentioned in the main review that it should be emphasized more that the theoretical guarantees only apply to exact posterior collapse. Otherwise, the paper seems fine regarding limitations.

**Main Review:**

On the theoretical side, the main contribution seems to be formalizing the definitions for posterior collapse and latent variable non-identifiability, and having the idea to study posterior collapse from this perspective. Otherwise, Theorem 1 in itself is, as the paper mentions, straightforward.

The limitation is, obviously, that the definitions and the theorem assume exactness. Approximate posterior collapse can be just as harmful, and practically indistinguishable from exact posterior collapse. The paper acknowledges some of this in lines 155-161. Nevertheless, I feel that this limitation should be emphasized earlier and more clearly, e.g., in the paragraph 37-41 or somewhere else in the introduction. When people talk about posterior collapse, my guess is that they mean approximate posterior collapse too (at least with “small epsilon”).

The Brenier maps idea is interesting. As before, it only ensures that no exact posterior collapse takes place, but the experiments seem to do a good job of showing that in practice the technique prevents approximate posterior collapse too.

On the empirical side:

In figure 1, (c) and (d), why is GMVAE trained for fewer epochs that ID-GMVAE? My guess is that it converges by that point, but I feel that something should be said explicitly, otherwise the figure looks incomplete. Also, in figure 1, (d), the legend hides a part of the data; I’m sure there’s some way to make it fit, or to move the legend somewhere else.

The results in Table 1 look solid, and they confirm the main claims of the paper.

Typo: 223-224: “is is”.

----

UPDATE: I acknowledge reading the authors' response as well as the other reviews. I maintain my original score.

**Time Spent Reviewing:**

4

---

> ### Author Response · Authors · 2021-08-10
> **Thank you very much for your review**
>
>
> Thank you very much for your review. We very much appreciate your
> positive comments on the paper. Below we address your specific
> comments.
>
>
> > The limitation is, obviously, that the definitions and the theorem
> > assume exactness. Approximate posterior collapse can be just as
> > harmful, and practically indistinguishable from exact posterior
> > collapse. The paper acknowledges some of this in lines 155-161.
> > Nevertheless, I feel that this limitation should be emphasized
> > earlier and more clearly, e.g., in the paragraph 37-41 or somewhere
> > else in the introduction. When people talk about posterior collapse,
> > my guess is that they mean approximate posterior collapse too (at
> > least with "small epsilon").
>
>
> Thank you very much for the comment and the suggestion. We have added
> this discussion on the exact vs approximate posterior collapse to the
> introduction to highlight the limitations of the analysis.
>
>
> > In figure 1, (c) and (d), why is GMVAE trained for fewer epochs that
> > ID-GMVAE? My guess is that it converges by that point, but I feel
> > that something should be said explicitly, otherwise the figure looks
> > incomplete. Also, in figure 1, (d), the legend hides a part of the
> > data; I’m sure there’s some way to make it fit, or to move the
> > legend somewhere else.
>
>
> Yes, we have added a discussion to explicitly say that the GMVAE is
> trained for fewer epochs because it has converged by that point. We
> have also moved the legend of Fig. 1d.
>
>
> > Typo: 223-224: “is is”.
>
> Thank you! We fixed this.

---

### Official Review · Reviewer_qzPP · 2021-07-15

**Rating:** 6
**Confidence:** 3

**Summary:**

The authors propose a parametrization of the VAE decoder which ensures that it is injective in the latent space, for a fixed observation. They show that such an injective decoder implies "latent identifiability", which in turn is equivalent to he VAE avoiding posterior collapse.

**Limitations And Societal Impact:**

The discussion at the end of the paper is short, and serves more as a conclusion and a recap of the contribution than the limits of their method.

**Main Review:**

The work provided by the authors is significant. The paper proves an equivalence between latent variable unidentifiability and posterior collapse. To prevent the latter, it is enough to use injective decoders in a VAE, which would result in latent variable identifiability.
Their important main theoretical result clearly shows that posterior collapse is a property of the model and/or data rather than the optimization of inference scheme.
The authors propose to use Brenier maps, along with an injective linear map to match dimensions, to define the decoder in their VAE.
Finally, this is accompanied by an empirical study that validates the capabilities of their model, and their superior performance in avoiding posterior collapse when compared to baseline methods.

However, the same can't said about the clarity and quality of writing, mainly due to bad choices when it comes to notations:

1- The examples in section 2.2 are not clear. This mostly ties to bad notations: from equation on line 166, $\theta$ seems to only parameterize f, but the examples (when referring to true parameters in line 176 for e.g.) use a $\theta$ that includes both $\mu$ and $\Sigma$.

2- The authors use bold variables $\mathbf{x}$ to refer to a dataset, and non-bold variables $x_i$ to refer to the $i$-th element of the dataset, but then use a bold $\mathbf{z}_d$ in line 318 to refer to some variable that is not even defined, and I'm still not sure what it means. I don't understand why standard notations (bold letters for vectors, non-bold letters for scalars) where not used, which will tremendously improve the clarity of the paper.

3- What does the notation $x \sim EF(u; v)$ mean? I can understand that $v$ is the vector of parameters for the exponential family, but what is $u$ here?  The use of this notation is also inconsistent, c.f. eq (7) and eq (1) for e.g.

4- what is the nature of $h_l$ in equation (6)? Is it a function from $R^d$ to $R$, or a pointwise function from $R^d$ to $R^d$? surely the $z_l$s in this equations are vectors (again, use of bold letters would have helped here..) since they are multiplied by matrices $W_l$ and $A_l$, yet they are equal to $h_{l-1}(...)$, which implies that it can't be a function from $R^d$ to $R$. But then, the authors say that $h_l$ is non-decreasing, which surely implies that it is a function between 2 ordered sets ($R^d$ is not an ordered set). All of this confusion can be solved by specifying that $h_l$ is a pointwise function for e.g., or by using better notations.

5- What are $\mathcal{Z}$ and $\mathcal{X}$? These are not defined anywhere in the paper.

6- inconsistent use of $w$ and $z$ between equation (9) and unnumbered equation after line 166.

7- The proposed model is called identifiable VAE, yet this terminology was used in one of the references they cite [23].

8- in paragraph on line 149, they argue that existing approaches can help mitigate posterior collapse. If we take $\beta$-VAE for e.g., this model seeks disentangled representation by putting a high weight, via the hyperparameter $\beta$, on the KL term of the ELBO. This pushes the posterior to match the factorized prior, thus achieving "disentanglement". This leads to posterior collapse, which is opposite to what is claimed in this section, unless I misunderstood something.

While I think that the contribution of the authors is worthwhile, the current presentation is not up to the standards of NeurIPS. But I'm happy to increase my score if the authors provide solutions to my concerns above

**Time Spent Reviewing:**

8 hours

---

> ### Author Response · Authors · 2021-08-10
> **Thank you very much for your review**
>
>
> Thank you very much for your review. We very much appreciate your
> positive comments on the technical approach of our paper. Below we
> address the specific concerns you raised about the notations and
> writing.
>
>
> > 1- The examples in section 2.2 are not clear. This mostly ties to
> > bad notations: from equation on line 166, \theta seems to only
> > parameterize f, but the examples (when referring to true parameters
> > in line 176 for e.g.) use a \theta that includes both \mu and
> > \Sigma.
>
> Thank you for pointing this out. We meant to use \theta to represent
> all parameters, including \mu, \Sigma, and the neural network f. We
> have fixed this notation:
>
>
> p(z_i) = Categorical(1/K),
>
> p(w_i|z_i; \mu, \Sigma) = N(\mu_z, \Sigma_z),
>
> p(x_i|w_i; f, \sigma) = N(f(z_i), sigma^2 * I_m),
>
> where the parameters include \theta = (\mu, \Sigma, f, \sigma^2).
>
>
> > 2- The authors use bold variables x to refer to a dataset, and
> > non-bold variables x_i to refer to the i-th element of the dataset,
> > but then use a bold z_d in line 318 to refer to some variable that
> > is not even defined, and I'm still not sure what it means. I don't
> > understand why standard notations (bold letters for vectors,
> > non-bold letters for scalars) where not used, which will
> > tremendously improve the clarity of the paper.
>
> Thank you very much for the excellent catch. Yes, the z_d stands for
> the d-th dimension of the latent variable z for all the n data points,
> z_d = (z_{1d}, ..., z_{nd}). In calculating AU, we follow Burda et al.
> (2015) to calculate the posterior mean, (E[z_{1d} | x_1], ...,
> E[z_{nd} | x_n]) for all data points, and calculate the sample
> variance of E[z_{id} | x_i] from this vector. We clarified this
> notation, and followed the standard notations in the revision to use
> bold letters for vectors and non-bold letters for scalars.
>
>
> > 3- What does the notation x ~ EF (u; v) mean? I can understand that
> > v is the vector of parameters for the exponential family, but what
> > is u here? The use of this notation is also inconsistent, c.f. eq
> > (7) and eq (1) for e.g.
>
> Thank you very much for the excellent catch. There is a typo in eq
> (7). We should use x ~ EF(x | u, v) to indicate x follows an
> exponential family distribution with parameters u and v. So eq (7)
> should be
>
> z_i ~ p(z_i),
>
> x_i | z_i ~ EF(x_i | g_{2,\theta}(beta^\top g{1,\theta)(z_i)), \lambda_\theta),
>
> where x_i follows an exponential family distribution with parameters
> g_{2,\theta}(beta^\top g{1,\theta)(z_i)) and \lambda_\theta.
>
> Accordingly, eq (1) should be
>
> z_i ~ p(z_i),
>
> x_i | z_i ~ p(x_i | z_i; theta) = EF(x_i | f_\theta(z_i)),
>
> where x_i follows an exponential family distribution with parameters
> f_\theta(z_i).
>
>
> > 4- what is the nature of h_l in equation (6)? Is it a function from
> > R_d to R, or a pointwise function from R_d to R_d? surely the z_l's
> > in this equations are vectors (again, use of bold letters would have
> > helped here..) since they are multiplied by matrices W_l and A_l,
> > yet they are equal to h_{l-1}(...), which implies that it can't be
> > a function from R_d to R. But then, the authors say that h_l is
> > non-decreasing, which surely implies that it is a function between 2
> > ordered sets (R_d is not an ordered set). All of this confusion can
> > be solved by specifying that h_l is a pointwise function for e.g.,
> > or by using better notations.
>
> Thank you for pointing this issue out. h_l denotes the entry-wise
> activation function at the layer l. It is a function from R -> R, and
> is applied element-wise to (W_l z_l + A_l u + b_l). We have clarified
> this notation in the paper. We have also fixed up this notation to be
> consistent with the convention of using bold letters for vectors and
> non-bold letters for scalars.
>
>
> > 5- What are \mathcal{Z} and \mathcal{X}? These are not defined
> > anywhere in the paper.
>
> Thank you for point it out. \mathcal{Z} and \mathcal{X} indicate the
> set of values the vector z_i and x_i can take. For example, if x_i is
> an m-dimensional binary vector, then \mathcal{X} = {0,1}^m. If z_i is
> a K-dimensional real-valued vector, then \mathcal{Z} = \mathbb{R}^d.
> We have clarified this notation in the paper.
>
> >
> > 6- inconsistent use of w and z between equation (9) and unnumbered
> > equation after line 166.
>
> Thank you for pointing this out. We have switched the use of w_i and
> z_i in Eq. (9), using z_i to always denote the variable for which we
> worry about collapse.
>
>
> z_i ~ Categorical(1/K),
>
> w_i|z_i ~ EF(w_i | \beta^\top_1 z_i, \gamma_\theta),
>
> x_i|w_i ~ EF(x_i | g_{2,\theta}(beta^\top g{1,\theta)(w_i)), \lambda_\theta).
>
>
> > 7- The proposed model is called identifiable VAE, yet this
> > terminology was used in one of the references they cite [23].
>
> Thank you for pointing out this issue. We have renamed our model as
> "latent-identifiable VAE" to avoid using existing terminologies.
>
>
> > 8- in paragraph on line 149, they argue that existing approaches can
> > help mitigate posterior collapse. If we take \beta-VAE for e.g.,
> > this model seeks disentangled representation by putting a high
> > weight, via the hyperparameter β , on the KL term of the ELBO. This
> > pushes the posterior to match the factorized prior, thus achieving
> > "disentanglement". This leads to posterior collapse, which is
> > opposite to what is claimed in this section, unless I misunderstood
> > something.
>
> Thank you for pointing this out. We would like to clarify that, by
> \beta-VAE, we meant its other use: putting a *low* weight on the KL term
> of the ELBO via the hyperparameter \beta (i.e. \beta < 1), as in Alemi
> et al. (2017). This is akin to the approaches of downweighting the
> prior to avoid posterior collapse (Razavi et al., 2019). A possible
> way of interpreting this use is that it helps find a \theta parameter
> under which the latent variable z becomes identifiable.
>
>
> Alemi, A., Poole, B., Fischer, I., Dillon, J., Saurous, R. A., & Murphy, K. (2018, July). Fixing a broken ELBO. In International Conference on Machine Learning (pp. 159-168). PMLR.
>
> Razavi, A., Oord, A. v. d., Poole, B., & Vinyals, O. (2019).
> Preventing posterior collapse with delta-VAEs. arXiv preprint
> arXiv:1901.03416.
>
>
> > Limitations And Societal Impact: The discussion at the end of the
> > paper is short, and serves more as a conclusion and a recap of the
> > contribution than the limits of their method.
>
> Thank you for pointing this out. We have added a more thorough
> discussion paragraph to restate the limitations, including the exact
> vs approximate posterior collapse we theoretically study (L155-161),
> the computational price we pay (L276-281), and the difference between
> latent variable non-identifiability and model non-identifiability
> (L364-372).

---

> > ### Comment · Reviewer_qzPP · 2021-08-25
> > **Thank you for your reply, mostly satisfied**
> >
> > Thank you for your reply!
> >
> > The authors have addressed most of my concerns.
> >
> > Before I update my score, there is one final point I'd like to make regarding the EF notation: I still think it is inconsistent, despite the changes you proposed.
> > In here:
> > $$x_i | z_i \sim EF(x_i | g_{2,\theta}(\beta^\top g_{1,\theta}(z_i)), \lambda_\theta)$$
> > the EF has two (vector) parameters, while in
> > $$x_i | z_i \sim p(x_i | z_i; \theta) = EF(x_i | f_\theta(z_i))$$
> > it has only one parameter. You can see why this is a source of confusion. The classical definition of an exponential family involves only one (vector) parameter so I' m not sure what the first definition means. Perhaps you were going for Gaussian distribution? In which case, I suggest saying that explicitly.

---

> > > ### Author Response · Authors · 2021-08-25
> > > **Thank you very much for the comment!**
> > >
> > > Thank you very much for the comment! We agree that this inconsistency could become a source of confusion. Thank you for pointing it out! We will fix the first definition to use only one (vector) parameter so that it is consistent with the later EF notations.

---

> > > > ### Comment · Reviewer_qzPP · 2021-08-30
> > > > **Increasing score**
> > > >
> > > > Thank you for your reply. I'm happy with the rebuttal and will increase my score to 6.

---

### Official Review · Reviewer_Fkyr · 2021-07-16

**Rating:** 6
**Confidence:** 4

**Summary:**

The authors investigate the reasoning behind the posterior collapse in VAEs and attribute it to non-identifiability. They propose an identifiable VAE that avoids the posterior collapse from happening.

**Main Review:**

General comments:
The paper is well-structured, and the motivation is well-defined. There is a need to understand the tendency of posterior collapse to ensure that latent variable models can scale to more significant problems. Therefore, it is highly relevant to the research community.

I like the motivation of using Brenier maps.

The caveat of this research is 1) that it has not been properly compared to previous research on the topic, e.g., Chen et al. 2017, and 2) that it has not been evaluated on state-of-the-art hierarchical VAE architectures, e.g., Maaloe 2019, Vahdat 2020, and Child 2021. Finally, I'm skeptical of the reasoning of Definition 1 in the paper (see comments below), which significantly influences the rest of the analysis.

With a revision considering points 1) and 2) I would change my decision.

More in-depth comments:
You state that a posterior collapse occurs when the inferred posterior, p(z|x; best_theta), equals the uninformed prior, p(z). You also state that "when posterior collapse occurs, it prevents the latent variable from providing a meaningful summary of the dataset". Is this, in fact, true? Isn't there cases for which a neural network model can fit a complex input space into a simple non-informative prior distribution, e.g., N(0,I)? You should support your argument.

In Eq. 3 you should introduce the variables, e.g., z'.

I find the argument on non-identifiability to be very similar to the bits-back coding and information preference proposed by Chen et al. 2017, in which they introduce the Variational Lossy Autoencoders. I think that the comparison should be stated much more clearly.

From the LVAE in Sonderby et al, 2016, and other research papers, they show that posterior collapse especially occur in hierarchical latent variable models. I think that the proposed method should be tested on these cases? Furthermore, what about the recent state-of-the-art results on very deep VAEs by Maaloe 2019, Vahdat 2020, and Child 2021?


**Time Spent Reviewing:**

1.5

---

> ### Author Response · Authors · 2021-08-10
> **Thank you very much for your review**
>
> Thank you very much for your review. We appreciate your positive
> comments on the motivation and relevance of our paper, as well as the
> use of Brenier map as a solution to posterior collapse. Below we
> respond to the specific concerns you raised.
>
>
> > "Comparisons to Chen et al. 2017 (bits-back coding). I find the
> > argument on non-identifiability to be very similar to the bits-back
> > coding and information preference proposed by Chen et al. 2017, in
> > which they introduce the Variational Lossy Autoencoders. I think
> > that the comparison should be stated much more clearly."
>
> Thank you for pointing out the connection to the bits-back coding and
> information preference proposed by Chen et al. 2017. This work
> provides a complementary view of posterior collapse than Chen et al.
> (2017).
>
> Chen et al. (2017) focuses on a setting where the exact posterior does
> not collapse. It explains how the *inexact* variational approximation
> can lead to inefficiency of coding in VAE, which could lead to
> posterior collapse due to "information preference."
>
> In contrast, our work shows how the exact posterior can collapse due
> to latent variable non-identifiability. Specifically, our work focuses
> on a setting when the variational approximation is exact. We show
> that, even when the variational approximation is *exact*, the
> posterior collapses when the latent variable is non-identifiable.
>
> That said, the two arguments are related in that the information
> preference argument is applicable when some form of ambiguity (e.g.
> non-identifiability) exists. In more detail, the information
> preference argument says: "information that can be modeled locally by
> decoding distribution p(x|z) without access to z will be encoded
> locally and only the remainder will be encoded in z." This argument
> rests on the premise that some information can be encoded *equally
> well* in the decoding distribution p(x|z) as in the latent variable z;
> it is a form of non-identifiability, though a different one than latent
> variable non-identifiability.
>
> We have clarified these connections in the revision.
>
> Finally, we have taken the suggestion to compare with variational
> lossy autoencoders (VLAE) in all the five image and text datasets
> studied in our empirical studies (Table 1). VLAEs are outperformed by
> the identifiable VAEs proposed in this work.
>
> Table: Performance of Variational Lossy Autoencoders (VLAE) (Higher is
> better.)
>
> ####Fashion MNIST
>
> | method | AU      | KL      | MI      | LL      |
>
> | VLAE (Chen et al., 2017)       | 0.3     | 0.4     | 1.3     | -250.6  |
>
> | IDGMVAE (this work)            | 1.0     | 1.6     | 2.6     | -242.3  |
>
> ####Omniglot
> | method | AU      | KL      | MI      | LL      |
>
> | VLAE (Chen et al., 2017)       | 0.2     | 0.4     | 1.1     | -852.9  |
>
> | IDGMVAE (this work)            | 1.0     | 1.7     | 7.5     | -820.3  |
>
> ####Synthetic Text
>
> | method | AU      | KL      | MI      | LL      |
>
> | VLAE (Chen et al., 2017)       | 0.3     | 0.1     | 0.1     | -40.6   |
>
> | IDSVAE (this work)             | 1.0     | 0.5     | 0.6     | -40.3   |
>
> ####Yahoo
> | method | AU      | KL      | MI      | LL      |
>
> | VLAE (Chen et al., 2017)       | 0.2     | 1.4     | 0.4     | -518.9  |
>
> | IDSVAE (this work)             | 0.9     | 9.2     | 1.4     | -517.5  |
>
>
> ####Yelp
> | method | AU      | KL      | MI      | LL      |
>
> | VLAE (Chen et al., 2017)       | 0.2     | 1.8     | 0.1     | -631.8  |
>
> | IDSVAE (this work)             | 0.9     | 9.3     | 1.2     | -631.2  |
>
>
> > "From the LVAE in Sonderby et al, 2016, and other research papers,
> > they show that posterior collapse especially occur in hierarchical
> > latent variable models. I think that the proposed method should be
> > tested on these cases? Furthermore, what about the recent
> > state-of-the-art results on very deep VAEs by Maaloe 2019, Vahdat
> > 2020, and Child 2021?"
>
> Thank you for suggesting the evaluation on hierarchical VAEs. The
> GMVAE in the empirical studies (Section 4.1) is one example of a
> two-layer hierarchical VAE; it has two layers of latent variables w_i
> and z_i:
>
> w_i ~ Categorical(1/K),
>
> z_i|w_i ~ Gaussian(g_1(w_i), exp(g_2(w_i) * I_d),
>
> x_i|z_i ~ Gaussian(f(z_i), sigma^2 * I_m),
>
> where f, g_1, g_2 are neural networks. Posterior inference is required
> for both layers of latent variables w_i and z_i.
>
> Consistent with the observation in Sonderby et al, 2016, posterior
> collapse indeed occurs in such hierarchical VAEs. The top row of Table
> 1 is the result for such hierarchical VAEs; it shows that classical
> VAEs will learn completely uninformative latent variables z_i's. In
> contrast, the IDVAE does not suffer from collapse at all.
>
> Finally, thank you for suggesting the comparison to Maaloe 2019,
> Vahdat 2020, and Child 2021. We were able to finish a comparison with
> Maaloe 2019 (bidirectional inference) and Vahdat 2020 (KL-balancing)
> in the GMVAE experiment during the rebuttal period.
>
> Table: Performance of Maaloe 2019 and Vahdat 2020 in GMVAE (Higher is
> better.)
>
> ####Fashion MNIST
>
> | method | AU      | KL      | MI      | LL      |
>
> | Bi-dir. inf. (Maaloe 2019)     | 0.3     | 0.3     | 1.2     | -252.6  |
>
> | KL-balancing (Vahdat 2020)     | 0.4     | 0.5     | 1.5     | -249.7  |
>
> | IDGMVAE (this work)            | 1.0     | 1.6     | 2.6     | -242.3  |
>
> ####Omniglot
>
> | method | AU      | KL      | MI      | LL      |
>
> | Bi-dir. inf. (Maaloe 2019)     | 0.3     | 0.6     | 1.1     | -852.1  |
>
> | KL-balancing (Vahdat 2020)     | 0.3     | 0.5     | 1.4     | -849.1  |
>
> | IDGMVAE (this work)            | 1.0     | 1.7     | 7.5     | -820.3  |
>
>
> > "Finally, I'm skeptical of the reasoning of Definition 1 in the
> > paper (see comments below), which significantly influences the rest
> > of the analysis. More in-depth comments: You state that a posterior
> > collapse occurs when the inferred posterior, p(z|x; best_\theta),
> > equals the uninformed prior, p(z). You also state that "when
> > posterior collapse occurs, it prevents the latent variable from
> > providing a meaningful summary of the dataset". Is this, in fact,
> > true? Isn't there cases for which a neural network model can fit a
> > complex input space into a simple non-informative prior
> > distribution, e.g., N(0,I)? You should support your argument."
>
> Thanks for your comment. We would like to clarify that we were
> referring to the use of latent variable posteriors for representation
> learning (Burgess et al, 2018; Tomczak and Welling, 2018) when we said
> "posterior collapse prevents the latent variable from providing a
> _meaningful summary_ of the dataset."
>
> Indeed, when posterior collapses, the variational autoencoder can
> still model the data well; posterior collapse does not compromise the
> effectiveness of variational autoencoders for density estimation.
>
> However, posterior collapse could create issues if we use the
> posteriors for representation learning. That is, for each datapoint
> x_i, we use the posterior mean of its latent variable E[z_i | x_i;
> best_\theta] as the representation (Burgess et al, 2018; Tomczak and
> Welling, 2018). In more detail, when the posterior collapses, i.e.
>
> p(z_i | x_i) = p(z_i) for i=1, ..., n,
>
> all data points will have the same representation because they will
> all be equal to the mean of the prior,
>
> E[z_i | x_i] = E[z_i] = E[z_j] = E[z_j | x_j], for all i,j \in
> {1,...,n}.
>
> In this case, E[z_i | x_i] is no longer a useful representation that
> summarizes each data point x_i. We have clarified this point in the
> paper.
>
>
> Burgess, C. P., Higgins, I., Pal, A., Matthey, L., Watters, N., Desjardins, G., & Lerchner, A. (2018). Understanding disentangling in $\beta $-VAE. arXiv preprint arXiv:1804.03599.
>
> Tomczak, J., & Welling, M. (2018, March). VAE with a VampPrior. In International Conference on Artificial Intelligence and Statistics (pp. 1214-1223). PMLR.
>
>
> > In Eq. 3 you should introduce the variables, e.g., z'.
>
> Thank you for point it out. \tilde{z} and \tilde{z}' refer to two
> values the latent variable z can take. Eq. 3 needs to hold true for
> all  \tilde{z}, \tilde{z}' \in \mathcal{Z}, that is, any two values z
> can take.

---

> > ### Comment · Reviewer_Fkyr · 2021-08-28
> > **Thank you for a thorough response**
> >
> > I have chosen to revise my decision on the paper, based on your response.Thank you for the hard work

---

### Decision · Program_Chairs · 2021-09-27

**Decision:**

Accept (Poster)

**Comment:**

The authors focus on a particular notion of posterior collapse in VAEs, namely $p(z|x) = p(z), \forall x$, as well as a notion of injectivity of the decoder---namely that $p(\cdot|z) \neq p(\cdot|z')$. They show that in the *exact* sense, these two notions are equivalent. Moreover, they provide an architecture that ensures injectivity by using Brenier maps (i.e. derivatives of convex functions) and provide some empirical data that this architecture is better at avoiding posterior collapse.

The theory in the paper is quite simple and straightforward. The empirical results are fairly small-scale, but suggest this architecture may have promise to avoid posterior collapse.